# Novel Diagnostics and Therapeutics in Sepsis

**DOI:** 10.3390/biomedicines9030311

**Published:** 2021-03-18

**Authors:** Kieran Leong, Bhavita Gaglani, Ashish K. Khanna, Michael T. McCurdy

**Affiliations:** 1Division of Pulmonary & Critical Care, University of Maryland School of Medicine, Baltimore, MD 21201, USA; kleong@som.umaryland.edu; 2Department of Anesthesiology, Section on Critical Care Medicine, Wake Forest University Hospital, Winston-Salem, NC 27157, USA; bgaglani@wakehealth.edu (B.G.); ashish@or.org (A.K.K.); 3Department of Outcomes Research, Outcomes Research Consortium, Cleveland, OH 44195, USA

**Keywords:** sepsis, novel, diagnosis, diagnostics, therapy, therapeutic, treatment, management, biomarkers, innovation

## Abstract

Sepsis management demands early diagnosis and timely treatment that includes source control, antimicrobial therapy, and resuscitation. Currently employed diagnostic tools are ill-equipped to rapidly diagnose sepsis and isolate the offending pathogen, which limits the ability to offer targeted and lowest-toxicity treatment. Cutting edge diagnostics and therapeutics in development may improve time to diagnosis and address two broad management principles: (1) source control by removing the molecular infectious stimulus of sepsis, and (2) attenuation of the pathological immune response allowing the body to heal. This review addresses novel diagnostics and therapeutics and their role in the management of sepsis.

## 1. Introduction

Sepsis is a clinical syndrome resulting from a dysregulated inflammatory response to infection. In 2017, an estimated 48.9 million cases of sepsis accounted for 11 million (19.7%) deaths worldwide [1]. In 2013, in the United States alone, sepsis accounted for $24 billion in hospital expenditures with the financial burden rising significantly over the subsequent 5 years [2,3].

Delayed identification and incorrect treatment lead to worse outcomes, increased costs, and higher mortality [2]. Managing sepsis in the contemporary era revolves around early diagnosis, administration of antimicrobials, hemodynamic support with fluids and vasopressors, and source control via procedural drainage and removal of the inciting pathogen. While these interventions have led to a decrease in hospital mortality, significant shortcomings in early recognition and treatment of the underlying cause of sepsis remain: the biomolecular triggering and subsequent inciting of an uncontrolled inflammatory response [4]. Overreliance on culture data delays identification of an infectious etiology and increases the possibility of inappropriate antimicrobial selection. The downstream effects of delayed or inappropriate antimicrobials include emerging antimicrobial resistance, medication toxicity, adverse microbiome alterations, and ineffective therapy.

While source control on the macro scale is important, the trigger of the pathological inflammatory cascade in sepsis ultimately occurs at the molecular level. The complex interaction between infectious molecules and the immune system is often overlooked in present-day management of sepsis. Innovations that prevent or attenuate this pathological interaction, as well as novel supportive therapies that provide time for patients to recover, are essential to improve outcomes in sepsis. This review describes promising novel diagnostics and therapeutics for sepsis management.

## 2. Materials and Methods

A PubMed search was performed using the search terms “biomarkers, novel, diagnostic, therapeutic, sepsis” from November 2019 to November 2020. Articles were then screened manually for relevance and those with data with clinical applicability in humans, as defined by previous use in human subjects, were included. A total of 658 identified articles were manually reviewed, and a total of 57 articles were included. Further searches were used to augment knowledge in topics found during the original search and on novel technologies identified within the past year.

## 3. Novel Diagnostics in Sepsis

According to the Third International Consensus Definitions for Sepsis and Septic Shock (Sepsis-3), sepsis is defined as life-threatening organ dysfunction caused by the dysregulated host response to infection [5,6,7]. Early identification and diagnosis are essential, as prompt and appropriate treatment can improve survival [8]. Sepsis may result from any type of infection (most commonly bacterial) that affects the body (most commonly the lungs or urinary tract). In contrast, viral sepsis is caused by a viral infection (e.g., influenza) that also carries the potential for superimposed bacterial infection. The year 2020 highlighted the devastating impact of virally-mediated sepsis, triggered by severe acute respiratory syndrome coronavirus 2. Our review focuses mainly on contemporary novel diagnostics and therapeutics in bacterial sepsis identified in the literature search covering a one-year period (November 2019–November 2020).

Limited resources are currently available to aid in early diagnosis of sepsis. Though blood cultures can occasionally identify the responsible pathogen and direct later antimicrobial therapy, their inability to yield timely results limits their role in the initial diagnostic process. Several molecular approaches have been developed in order to improve conventional culture-based identification, including PCR and matrix-assisted laser desorption ionization–time of flight (MALDI–TOF) mass spectrometry. Although MALDI–TOF may decrease the time to result to as early as two and half hours once the blood cultures become positive [9,10], a broader clinical evaluation of this approach is still missing. Recent data suggest that transcriptomic profiling by multiplexed quantitative PCR (qPCR) and metabolite detection by liquid chromatography-tandem mass spectrometry (LC-MS/MS) have potential in the clinical development of diagnostic tests capable of overcoming the limitations of single molecules to differentiate between infectious and noninfectious causes of systemic inflammation [11].

Various clinical scoring systems, such as the sequential organ failure assessment (SOFA) score, exist to assist with the diagnosis of sepsis [5,7,12]. Although SOFA is one of many such tools, may help identify patients with increased risk of death, and is utilized in the current sepsis definitions [5], it is by no means specific for infection or sepsis [3].

Some biomarkers, such as C-reactive protein (CRP) and procalcitonin (PCT), have been widely used as an acute phase reactant in critically ill patients, but their diagnostic and prognostic values for sepsis are limited [13,14,15]. In a retrospective cohort study in critically ill patients fulfilling the Sepsis-3 criteria, the diagnostic accuracies of PCT and CRP insufficiently predicted proven infection, with no difference in decrease in both markers in 28-day survivors and nonsurvivors [16]. However, the multicenter, open-label, Procalcitonin-guided Antimicrobial Therapy to Reduce Long-Term Sequelae of Infections (PROGRESS) trial in 266 patients meeting Sepsis-3 criteria demonstrated that PCT-guided therapy, as compared to standard care, yielded a significantly reduced mortality [17]. With more than 100 biomarkers already described and proposed for sepsis [18], defining which marker may be useful to optimize diagnostics and therapeutic strategies remains a challenge.

A recent comprehensive review identified 5367 studies investigating the use of biomarkers in relation to sepsis [18], with a total of 80 new individual biomarkers emerging over the past decade. Of these 80, a mean of 21 biomarkers were assessed specifically for the diagnosis of sepsis in basic research studies, clinical studies, and studies combining both approaches. We attempted to categorize the various biomarkers according to pathophysiological roles (Figure 1), although for many, identifying a single clear role was not possible. While many studies have validated a multibiomarker-based risk model that estimates mortality probability in adults with septic shock [19], we focused on clinical studies in adults over the past year that compared biomarkers in different sepsis-related pathways. Table 1 summarizes different novel biomarkers reviewed below in detail.

### 3.1. Novel Innate Response Biomarkers

#### 3.1.1. PAMPS and DAMPS

Sepsis begins with the activation of an innate immune response mediated by the detection of damage-associated molecular patterns (DAMPs) or pathogen-associated molecular patterns (PAMPs) by pattern-recognition receptors (PRRs) on host cells. PAMPs are unique motifs found on microbes that are recognized by PRRs, and allow the innate immune system to distinguish self from non-self, while DAMPs are a sign that there is damage to the host. Both PAMPs and DAMPs function as “molecular warnings” that activate circulating and tissue-resident immune cells [20]. An FcMBL-based PAMP blood assay may detect the presence of pathogens and pathogenic material from the blood or localized to specific organ sites [20,21]. The use of multiplexed detection of PAMPs is currently being investigated and may revolutionize point-of-care diagnostics for a broad range of diseases and conditions.

#### 3.1.2. Calprotectin

Calprotectin is one of the most abundant proteins in the neutrophil cytosol, released from neutrophils reacting to bacterial infections. This calcium-binding protein, which consists of two subunits S100A8 and S100A9, typically increase within hours in response to bacteria or endotoxin [22]. In a prospective observational study, patients with sepsis exhibited significantly elevated serum calprotectin concentrations whereas PCT concentrations did not distinguish those with sepsis from those without [23]. Moreover, calprotectin levels directly correlated with 30-day mortality. Others have also demonstrated calprotectin’s superiority (area under the curve (AUC) 0.775, 95% confidence interval (CI) 0.667–0.861) to PCT (AUC 0.736, 95% CI 0.625–0.829) to differentiate bacterial pneumonia from viral pneumonia [24]. Several studies have highlighted the ability of calprotectin to identify bacterial infection. Further studies will likely address the clinical use of calprotectin to guide the initiation and de-escalation of antibiotic therapy in septic patients.

### 3.2. Novel Cytokine/Chemokine Biomarkers

#### 3.2.1. Interleukin 6 (IL-6)

Markers currently in use, such as CRP, PCT, and interleukin 6 (IL-6), were repeatedly found to be compared with each other in the literature review to justify its superiority in diagnosis of sepsis. In one study of patients with sepsis and septic shock diagnosed according to Sepsis-3, serum IL-6 could discriminate sepsis (area under the curve (AUC), 0.83–0.94, *p*  < 0.001, 80.4% sensitivity, 88.9% specificity) from controls; however, for distinguishing septic shock (AUC, 0.71–0.89, 76.1% sensitivity, 78.4% specificity) from sepsis, optimal cutoff value was similar to pentraxin 3 (PT3) and PCT [25]. In a Cochrane Database of Systematic Reviews analysis of 23 studies containing 4192 patients, the heterogeneity of existing studies assessing the diagnostic potential of IL-6 precluded the ability to calculate diagnostic accuracy estimates, leading the authors to conclude that further studies are required to thoroughly investigate the topic [26]. While infection by the severe acute respiratory syndrome-associated coronavirus (SARS-CoV-2) induces dose-dependent IL-6 production from bronchial epithelial cells [27], the role of IL-6 to detect sepsis, particularly in viral infection, remains inconclusive. Further studies should elucidate the sensitivity and specificity of a plasma inflammatory signature consisting of IL-6 to distinguish between viral and bacterial infections.

#### 3.2.2. Monocyte Chemoattractant Protein 1 (MCP1)

Often induced by oxidative stress, cytokines, or growth factors, cells such as endothelial cells and monocytes secrete the cytokine monocyte chemoattractant protein 1 (MCP-1), also known as C-C motif chemokine ligand 2 (CCL2), to initiate the inflammatory cascade [28]. Serum levels of MCP-1a directly correlate with sepsis severity [29]. In fact, MCP-1 levels may help discriminate septic shock from healthy and postoperative controls, with MCP-1 predicting 28-day mortality (AUC 0.763). Moreover, plasma MCP-1 levels directly correlate with other cytokine levels (TNF-α, IL-6, IL-8, and IL-10) in the setting of sepsis [30,31].

#### 3.2.3. Pentraxin 3 (PTX3)

Pentraxin 3 (PTX3) plays a role in the early phase of inflammation by activating the classical complement pathway and facilitating pathogen recognition by macrophages and dendritic cells [32]. A prospective study of 101 patients compared sepsis and healthy groups by measuring plasma PTX-3, MCP-1, and angiopoietin (Ang)-2 levels [25] on the first day of sepsis onset. All of these biomarkers were significantly increased in the sepsis group compared to the healthy group, and PTX-3 had the highest AUC of 0.798 (95% CI 0.666–0.921, *p* < 0.0001) for predicting septic shock. Another prospective single-centered study showed that serum PTX3 levels could identify both the diagnosis and severity of sepsis with AUC of 0.84 for PTX3 (95% CI, 0.95–0.99; *p* < 0.001) with optimal cut-off values of 15.10 ng/mL (sensitivity, 92.6%; specificity, 97.4%; *p* < 0.001)., suggesting that PTX3 has a diagnostic value comparable to that of IL-6 in sepsis and septic shock [30].

#### 3.2.4. sTNFR1

In addition to IL-6, early proinflammatory cytokines, such as TNF-α, IL-1, and IL-8 were studied to determine if their levels correlated with mortality from sepsis and septic shock, which are traditionally considered a consequence of an exacerbated early innate immune response [33]. In the case of TNF, two membrane-bound TNF receptors, TNFR1 and TNFR2, are released into the circulation to regulate inflammation. Cleavage of the extracellular portion of these receptors produces soluble molecules (sTNFR1 and sTNFR2) in the blood that retain the ability to bind TNF and inhibit its acute activity [34]. To characterize disorders in the innate immune response, the TLR4 receptor signaling pathway, including its effectors of pro- and anti-inflammatory cytokines (IL-1Ra, TNF-α, sTNFR1, IL-6, IL-10, sTLR4), was assessed in 163 severely ill ICU patients with infections [35]. The immune response greatly differed between severely ill patients with and without infections, and patients with infections exhibited concentrations of sTNFR1 even greater than those found in uninfected patients who died from their illness (AUC 0.686 and cut-off point = 24.841 pg/mL). In addition to its diagnostic value for severe infections, this protein exhibits prognostic value as well.

### 3.3. Novel Receptor Biomarkers

#### 3.3.1. Soluble Urokinase-Type Plasminogen Activator Receptor

The concentration of soluble urokinase-type plasminogen activator receptor (suPAR), a membrane-bound receptor widely found in blood and body fluids, directly correlates with immune system activity including cell adhesion, migration, chemotaxis, proteolysis, immune activation, tissue remodeling, invasion, and signal transduction [36].

Systematic reviews published previously indicate that suPAR has a moderate diagnostic value for bacterial infection or systemic inflammation with AUC of 0.82 [37]. Furthermore, in a recent systematic review and meta-analysis of 30 studies involving 6906 patients [38], suPAR and PCT demonstrated similar diagnostic accuracy for sepsis. The pooled sensitivity of suPAR in diagnosing sepsis was 0.76 (95% CI, 0.63–0.86; *I*^2^ = 93.71%, *Q* = 254.49 (*p* < 0.01)) and the specificity was 0.78. In addition, AUC for differentiating sepsis from systemic inflammatory response syndrome (SIRS) was 0.81 (95% CI, 0.77–0.84), and the sensitivity and specificity were 0.67 (95% CI, 0.58–0.76) and 0.82 (95% CI, 0.73–0.88), respectively [38]. Given its lack of sensitivity, further investigation is required to evaluate whether using of suPAR in combination with other biomarkers can improve diagnostic efficacy.

#### 3.3.2. Presepsin

Presepsin, a soluble subtype of CD14 (sCD14), is related to bacterial phagocytosis and lysosomal cleavage of microorganisms and is an emerging biomarker of infection and systemic inflammation [39,40]. In a 2019 systematic review and meta-analysis that included 19 observational studies with 3012 patients demonstrated a pooled sensitivity and specificity of 0.84 (95% CI, 0.80–0.88) and 0.73 (95% CI, 0.61–0.82) for presepsin and 0.80 (95% CI, 0.75–0.84) and 0.75 (95% CI, 0.67–0.81) for PCT to diagnose sepsis. [40]. The meta-analysis demonstrated the relatively equivalent performance of both presepsin and PCT, with AUCs of 0.87 and 0.84, respectively [40]. They concluded that either biomarker could be used in combination with other biomarkers as a potential diagnostic approach. Interestingly, in patients with severe acute kidney injury, the accuracy of the diagnosis of sepsis with procalcitonin (AUC 0.946) was found to be significantly higher than that for presepsin (0.768, *p* < 0.001) [41]. Overall, insufficient evidence supports the accuracy of presepsin compared with traditional biomarkers, such as procalcitonin or CRP to diagnose sepsis.

#### 3.3.3. CD64

CD64 is a high-affinity immunoglobulin Fc γ receptor expressed on monocytes, eosinophils, and neutrophils and responds to infection or exposure to endotoxins within a few hours [42]. In a 2019 meta-analysis of 14 studies and 2,471 patients [43], CD64 bettered CRP and PCT for diagnosing sepsis, as the area under the summary receiver operating characteristic (SROC) curve was larger for neutrophil CD64 than for CRP (0.89 (95% CI 0.87–0.92) vs. 0.84 (95% CI 0.80–0.88), *p* < 0.05) or PCT (0.89 (95% CI 0.84–0.95) vs. 0.84 (95% CI 0.79–0.89), *p* < 0.05). Utilizing CD64 in clinical practice, however, is challenging due to the need to measure it using flow cytometry. A novel smartphone-based technique, using a smartphone-imaged microfluidic biochip, demonstrated the feasibility of measuring CD64 in 37 specimens from eight patients, highlighting the potential to integrate such assessments into bedside patient care [44]. Prior studies suggested the potential of the CD64 index, in combination with more sensitive markers, to be utilized for medical ICU patients [45].

#### 3.3.4. sTREM-1

Soluble triggering receptors expressed on myeloid cells (sTREM-1) are mainly distributed on the surface of polymorphonuclear cells and mature monocytes and are up-regulated by bacterial lipopolysaccharides (LPS). Engagement of TREM-1 triggers secretion of IL-8, monocyte chemotactic protein-1, and TNF-α and induces neutrophil degranulation [46]. In a 2020 meta-analysis of 19 studies involving 2418 patients, the ability of sTREM-1 to diagnose sepsis had a pooled sensitivity and specificity of 0.82 (95% CI, 0.73–0.89) and 0.81 (95% CI, 0.75–0.86), respectively [47]. In adult populations, sTREM-1 levels in septic patients with a positive culture were significantly higher in nonsurvivors compared to survivors, but they failed to have any value in culture-negative septic patients [48]. Another study suggested that both CRP and IL-6 more accurately identified severe sepsis and septic shock than sTREM-1, although sTREM-1 did outperform PCT in diagnosing severe sepsis [48,49]. While some of these differences in diagnostic capabilities depended on which definition of sepsis was used, sample sizes in these studies were small and many of the studies questioned whether sTREM-1 has any clinical value.

#### 3.3.5. Toll-Like Receptor 4 (TLR 4)

Toll-like receptors (TLR) are the receptors involved in the induction of inflammatory genes [20,50]. Among TLRs, TLR4 can recognize LPS, other PAMPs, and DAMPs at the cell surface [51], whereas TLR3, TLR7, TLR8, and TLR9 are exclusively expressed in endosomal compartments and recognize viral components [50]. The discovery of TLRs in humans, and the early recognition of TLR-4 as the receptor that signals LPS bioactivity were major breakthroughs not only in the field of immunology but also in sepsis [52]. Despite a lack of larger studies demonstrating an association between TLR4 and biomarkers like PCT and CRP in diagnosis of sepsis, results of a recent study conclude the possibility of using TLR2 and TLR4 expression to determine the severity of sepsis as a diagnostic biomarker (*p* < 0.05) [53]. Additionally, some data support a link between TLR4 signaling and pathological inflammation during infection by SARS-CoV-2, the virus causing the disease Covid-19 [54]. Such findings may help identify methods of targeting TLR4-mediated inflammation to develop therapeutic approaches to sepsis.

#### 3.3.6. Programmed Death-1 (PD-1) Receptor

The programmed death-1 (PD-1) receptor, an inducible coinhibitory cell-surface protein expressed in T and B cells, is important for establishing immune tolerance [55]. PD-1 activation, along with its ligands PDL1 and PDL2, downregulates T cell activation to alter the balance between immune tolerance and immune-mediated organ damage. Prior studies demonstrated that regulatory T cells in patients with severe sepsis and septic shock exhibited greater expression of PD-1 [56]. In a recent prospective observational cohort study, the degree of NK cell expression of four co signaling molecules (PD-1, CD28, PD-L1, and CD86) directly correlated with sequential organ failure assessment (SOFA) scores [57]. While the percentage of PD-L1^+^ NK cells and SOFA scores were independent risk factors for 28-day mortality, the AUC of the percentage of PD-L1^+^ NK cells, SOFA score, and their combination were 0.655 (0.559–0.742), 0.727 (0.635–0.807), and 0.808 (0.723–0.876), respectively. The AUC of the combination model best predicted 28-day mortality (all *p* < 0.05) [57]. The ability of PD-1 to serve as a novel prognostic biomarker for mortality may further enhance the predictive capacity of the SOFA score in septic patients.

### 3.4. Novel Microcirculation-Related Biomarkers

#### 3.4.1. Angiopoietin-1 (Ang-1) and Angiopoietin 2 (Ang-2)

The angiopoietin (Ang)-Tie system helps control vascular endothelial cell responses during sepsis [58]. By regulating the Ang family, bacterial endotoxin affects the function of vascular endothelial cells, with Ang-1 inhibiting vascular permeability and activating the Tie-2 receptor and Ang-2 promoting vascular leakage by blocking the Tie-2 receptor [58,59,60], opposing each other’s actions. In several clinical studies of sepsis, both a high level of Ang-2 and a low level of Ang-1 or high Ang-2/Ang-1 and low Ang-1/Ang-2 ratios have been associated with poor clinical outcomes, organ dysfunction, and adverse outcomes in sepsis, including predicting the severity of acute respiratory distress syndrome (ARDS) [60,61]. In one small study utilizing the Sepsis-3 criteria in patients with sepsis and septic shock, those with septic shock demonstrated significantly elevated plasma levels of PTX3, MCP1, and Ang-2 and low levels of Ang-1 [30], and Ang2 levels in sepsis patients were significantly higher than in those patients without sepsis (*p* < 0.05), with AUC 0.631 (95% CI 0.464–0.799, *p* = 0.1288) for Ang2 expression in septic shock patients. In addition, Ang-2 directly correlated with coagulation and fibrinolysis indices, suggesting development of coagulopathy in patients with sepsis and septic shock with higher Ang 2 levels [30,61,62]. Using a preclinical study in mouse models of sepsis or acute lung injury, Ang-2 inhibition or Ang-1/Tie2 axis stimulation with Ang-2 neutralizing antibodies or an Ang-2-targeted short interfering RNA decreased mortality and the incidence of multiple-organ dysfunction syndrome [63]. Substantial opportunity exists surrounding the potential clinical role for angiopoietin as a sepsis biomarker for both diagnostic and prognostic purposes, as well as its role for targeted sepsis therapies.

#### 3.4.2. Adrenomedullin (ADM) and Pro-Adrenomedullin (ProADM)

Adrenomedullin (ADM), a 52-amino acid peptide produced mainly in endothelial cells and vascular smooth muscle cells, is secreted by various tissues. By mediating vasodilation as an autocrine/paracrine vasoactivator, ADM helps regulate systemic circulation. Because circulating ADM is quickly degraded and cleared from the blood, its levels are difficult to detect using a standard immunoassay. Levels of the more stable mid-regional fragment of pro-adrenomedullin (MR-proADM), comprised of amino acids 45–92, directly reflect the levels of the active ADM peptide and has been studied as biomarker in sepsis and septic shock [64,65]. Based on a prospective single-centered study, a combination test of PCT and MR-proADM may represent an effective tool in sepsis diagnosis and prognosis. Furthermore, MR-proADM serves as a marker of organ dysfunction and has a turnaround time of about 30 min. Recently, a double monoclonal sandwich immunoassay demonstrated its ability to measure C-terminally amidated biologically active ADM (bio-ADM) [66]. The AdrenOSS-1 study found that bio-ADM levels were higher in septic shock patients than in sepsis patients. In addition, the return of bio-ADM levels to normal values (<70 pg/mL), measured at 48 h after admission, correlated with decreased 28-day mortality and improved cardiovascular function [67]. Levels of bio-ADM also predicted 30-day mortality similar to the SOFA score (AUC 0.827 vs. 0.830) [68]. In light of these encouraging data, validating the optimal serum levels of bio-ADM for clinical use will be important before its use implementation as a sepsis biomarker.

### 3.5. Novel Biomarkers of Organ Dysfunction in Sepsis

#### 3.5.1. MicroRNA (miRNA)

MicroRNAs (miRNA), a class of small, non-coding RNAs, post-transcriptionally regulate up to 60% of protein encoding genes. Emerging evidence suggests that miRNAs are key mediators of the host response to infection, predominantly by regulating proteins involved in innate and adaptive immune pathways. The miRNA characteristics of the host-pathogen interactions of more than 50 different bacterial and viral infections have been described [69]. Although confirming the functional nature of such associations is difficult, such studies highlight the important role of miRNA in immunity and provide evidence that polymorphism in miRNA genes, at least for certain pathogens, could govern person-to-person variation in infection susceptibility [70,71,72,73]. To determine the accuracy of circulating miRNA as a biomarker for sepsis, one meta-analysis [71] assessed a total of 2337 patients in 14 studies of SIRS, 2 studies of local infections, and 14 studies of healthy controls. Circulating miRNA proved to be an accurate method to identify sepsis, with a pooled sensitivity and specificity of 0.80 (95% CI 0.75–0.83) and 0.85 (95% CI 0.80–0.89), respectively, and the AUC was 0.89 (95% CI 0.86–0.92). The use of differing sources of miRNA (e.g., whole blood, plasma, cerebrospinal fluid) tends to present the greatest challenge for studies attempting to validate miRNA levels identified in other studies. Other study differences that include distinct patient characteristics, pathogen types, study design methodologies, and presenting diseases (e.g., neurological disorder, malignancy, sepsis, aseptic inflammation) causing elevated miRNA levels [70,72,73], all contribute to the difficulty confirming the diagnostic accuracy of the test.

#### 3.5.2. Long Non-Coding RNAs (lncRNAs)

Long non-coding RNAs (lncRNAs) are a class of non-coding RNAs with transcripts of more than 200 nucleotides with limited protein-coding ability. Although many non-coding RNAs associated with inflammatory diseases, including sepsis, have been identified, their functions and mechanisms are not well known and are controversial. A prospective cohort study of 120 sepsis patients showed that lnc-MALAT1 accurately diagnosed sepsis (AUC 0.910) and predicted 28-day survival (AUC 0.886) better than the APACHE II score (AUC 0.868) and lactate levels (AUC 0.868) [74]. Further evaluation of the roles of non-coding RNAs in the pathogenesis of sepsis and its appropriate use in the clinical setting is warranted.

#### 3.5.3. Matrix Metalloproteinases (MMPs)

Matrix metalloproteinases (MMPs) and tissue inhibitors of metalloproteinases (TIMPs) are key mediators in the regulation of wound healing after internal injury [75]. To investigate the MMP-2, MMP-9, TIMP-1, TIMP-2 and IL-6 plasma levels in patients with severe sepsis and to examine their association with prognosis, the 37 patients on day 1 of severe sepsis and 37 healthy volunteers were enrolled [76], and the protein levels were measured by ELISA methods. The results indicate that levels of MMP-9, TIMP-1, TIMP-2 and IL-6 in septic patients were significantly higher compared to healthy controls (*p* < 0.001). Another study evaluating early circulating plasma levels of MMP-2, MMP-9, and their inhibitors TIMP-1 and TIMP-2 and their prognostic significance in critically ill patients on admission to the intensive care unit (ICU) [77] found that 30-day survivors had significantly lower plasma MMP-9 (odds ratio, OR 1.67 per 1 SD; 95% confidence interval, CI 1.10–2.53; *p* = 0.016) and TIMP-1 (OR 2.15 per 1 SD; 95% CI 1.27–3.64; *p* = 0.004) levels than nonsurvivors. More studies are needed to evaluate its diagnostic accuracy and correlation to the Sepsis-3 criteria.

### 3.6. Nanodiagnostics

Nanotechnology-based biosensors are a novel approach to diagnostics with improved sensitivity for biomarkers and shorter processing times without the requirement of specialized skills [78]. Biosensors are devices that generate signals in proportion to concentrations of analytes in biological samples [79], permitting the characterization of minuscule signals using a small number of samples from various bodily fluids. The development of nanosensors based on electrochemical, immunological, or magnetic principals provide highly sensitive, selective, and rapid detection of sepsis biomarkers such as PCT and CRP [78,79]. Nanodiagnostics could potentially serve as a diagnostic platform designed to monitor an array of key pro- and antiinflammatory biomarkers and monitor the pattern of an individual patient’s immune response to provide early targeted treatment [80,81]. However, further translational studies of nanotechnologies will be required to evaluate their clinical and cost-effectiveness against the current standard of care.

## 4. Therapeutics

The mainstay of therapy in sepsis revolves around two broad principles: (1) source control to remove the infectious stimulus and (2) resuscitation optimization to both attenuate the pathologic inflammatory response and provide end-organ support. Current source control therapies include antimicrobial administration and procedural interventions to reduce the pathogenic burden [82]. Unfortunately, these therapies incompletely address the integral role of infectious molecular triggers to incite and propagate the characteristic inflammatory cascade of sepsis that manifests itself to different degrees according to each patient’s unique immune system and biochemical milieu. Meanwhile, supportive care is often limited to the implementation and titration of therapies such as intravenous fluids, vasopressors, mechanical ventilation, and renal replacement therapy (RRT). Comprehensive application of the correct combination of the above therapies in a timely manner improves outcomes in sepsis. We discuss novel therapies (Table 2) that target the pathogen burden and those that target the host by attenuating the adverse effects of the molecular triggers of inflammation to support patients until recovery.

### 4.1. Pathogen-Directed Therapies

#### 4.1.1. Pathogen-Associated Molecular Pattern Removal Devices

Pathogen-associated molecular patterns (PAMPs) induce immune cells to release proinflammatory mediators that can trigger the dysregulated inflammatory response in sepsis. The attractive aspect of filters targeting PAMP removal, as opposed to implementing methods that target specific cytokine removal deactivation, is the ability to remove upstream triggers of inflammation in lieu of attempting to correct the downstream and less understood milieu of both good and bad inflammatory markers. Several promising devices remove PAMPs using extracorporeal hemofiltration devices, often in conjunction with either RRT or extracorporeal membrane oxygenation. Due to growing concerns about the consequences for multidrug-resistant (MDR) pathogens, the Defense Advanced Research Projects Agency (DARPA) has provided substantial investments to expand this field.

The GARNET device, created by BOA Biomedical, is an extracorporeal hemofiltration device that utilizes the Fc-mannose-binding lectin (FcMBL) attached to fibers within a hollow cartridge. FcMBL is created from the Fc portion of human immunoglobulin, which is attached to a carbohydrate recognition domain of mannose-binding lectin (MBL) [95]. MBL, a blood opsonin, can recognize and bind a wide range of PAMPs. FcMBL is capable of binding 85% of isolates from 97 of 112 (87%) pathogen species, including the most common pathogens responsible for sepsis, and a wide range of bacteria, parasites, and viral antigens [95]. A preclinical trial in rats showed a >90% decrease in bacterial load and improved survival after five hours of use [96]. Patients are currently being enrolled in a multicenter trial using the BOA Biomedical device [83].

The hemofiltration device Seraph-100, produced by ExThera, uses heparin sulfate-coated absorption beads within a cartridge to bind and sequester pathogens. In a case study, hemofiltration using this device decreased the bacterial load of Staphylococcus aureus. Following case studies demonstrating improved hemodynamics with its use, Seraph-100 was approved for use in the European Union and received emergency use authorization (EUA) for the treatment of severe Covid-19 by the Food and Drug Administration (FDA) [97].

Oxiris, owned by Baxter, is an adsorptive membrane with a negatively charged, microporous architecture created to remove cytokines and endotoxins during RRT. Its use in sepsis, when compared to historical controls, reduces SOFA scores by 37% at 48 h [98]. Due to its ability to reduce serum IL-6 levels, it received an EUA from the FDA for use in severely ill Covid-19 patients.

Another promising device, Cytosorb, is a hemofiltration device created with biocompatible polystyrene divinylbenzene copolymer beads. It clears both proinflammatory cytokines and PAMPs but is unable to clear endotoxins [98,99]. Its use decreased IL-6 levels and mortality in one observational study of septic patients on continuous RRT; however, the only RCT completed noted no significant change in IL-6 levels, SOFA scores, or mortality [98,100]. This may be due to the relatively short (6 h) daily therapy administered. It has received an EUA for use in the setting of Covid-19 accompanied by elevated cytokines.

Two other adsorption devices, Toraymyxin and Alteco LPS, specifically target endotoxin removal. Toraymyxin is approved for use in Japan to treat patients with gram-negative bacterial infection, however no randomized controlled trials (RCTs) have shown decreased mortality for endotoxin clearance alone [98]. The Alteco LPS device, however, demonstrated an improvement in patients’ hemodynamics [98].

While PAMP removal technology is promising and ongoing trials may demonstrate usefulness, FDA approval does not yet exist for its use in sepsis [92,100].

#### 4.1.2. Bacteriophages

Phage therapy has experienced renewed interest in recent years due to the emergence of MDR bacteria [84]. Bacteriophages are possible alternatives to antibiotics given their ability to cleave capsular polysaccharides on organisms such as *Klebsiella pneumoniae* [101]. Murine models of sepsis demonstrated that a single dose of the studied phage protected 80–100% of subjects against death [101]. Additionally, case reports document good outcomes resulting from the use of bacteriophages in human patients with recalcitrant and pan-resistant gram-negative bacteremia [102].

#### 4.1.3. Intravenous Immunoglobulin

Intravenous immunoglobulin (IVIG) is an infusion of pooled IgG immunoglobulins that targets several organisms or inflammatory conditions, including Guillain–Barre syndrome, immune thrombocytopenic purpura, and Kawasaki disease [100]. IVIG has been used in sepsis to both inhibit the inflammatory response and to opsonize the offending infectious agent. Data are inconclusive regarding its efficacy in sepsis and is currently not recommended for routine use [85,92]. However, a recent meta-analysis of IV IgM infusions showing reduced mortality in those with septic shock has renewed interest in this therapy [103].

#### 4.1.4. Targeted Monoclonal Antibodies

Direct antibacterial antibodies targeting *Pseudomonas aeruginosa* and *Staphylococcus aureus* are undergoing clinical trials [85]. Two monoclonal antibodies against *Staphylococcus aureus* toxin, suvratoxumab and AR-301, exist. Suvratoxumab reduces disease severity in mice and has undergone a phase 2 trial which showed a trend towards reduced incidence of *Staphylococcus aureus* pneumonia when used preemptively in high risk, mechanically ventilated ICU patients [104]. AR-301, in a phase 2 trial, showed a trend towards faster resolution of pneumonia in those with severe *Staphylococcus aureus* hospital-acquired pneumonia, ventilator-associated pneumonia, or community-acquired pneumonia [105]. A phase 3 trial is currently underway [85].

#### 4.1.5. Liposomes

Many bacteria secrete toxins that damage cellular structures. Artificial liposomes, which can bind and sequester these toxins [106], have demonstrated improved survival in mice with Streptococcus pneumonia septicemia [85]. When compared to controls, CAL02, one such agent, showed a reduction in organ dysfunction scores in 19 patients admitted to an intensive care unit for severe pneumococcal pneumonia [86].

#### 4.1.6. Alkaline Phosphatase

Sepsis-mediated acute kidney injury causes renal cell apoptosis and increases the risk of mortality and progression to end-stage kidney disease [85]. Alkaline phosphatase protects against renal inflammation by deactivating bacterial LPS [85]. A phase 2 study in patients with sepsis-induced acute kidney injury demonstrated a measurable improvement in renal function in those with shock [107]. Larger phase 2 studies also noted an improved creatinine clearance three to four weeks after randomization, as well as a statistically significant decrease in mortality [87].

#### 4.1.7. Antimicrobial Peptides

Antimicrobial peptides (AMPs) are small proteins which belong to the innate immune system and have potent antibacterial, antiviral, and antifungal activity [108,109]. They function in three distinct ways: 1) direct antimicrobial activity on target cell membrane, 2) antimicrobial activity via immune modulation, and 3) inhibition of bacterial intracellular function [105]. Concerns of concomitant host cytotoxicity due to minimal specificity exist, though recent evidence suggests that AMPs are functionally diverse in their targets [108,109]. Administration of AMPS concomitantly with antimicrobials may be synergistic and minimize patient toxicity [108,109]. AMPs are currently undergoing phase 3 clinical trials in severe sepsis [88].

#### 4.1.8. Nanoparticles

Nanoparticles comprise a rapidly growing field of research dedicated to sepsis therapeutics. These engineered therapies target specific microbes to both enhance the potency of systemic antimicrobials and minimize their side effects [78,89]. By directly targeting infectious organisms and neutralizing endotoxin, they also may reduce emerging antimicrobial resistance [89]. For example, the commercially available liposomal formulation of amphotericin B is a drug delivery nanosystem incorporated into a lipid bilayer that is only released on exposure to the targeted fungus, thus reducing the toxicity traditionally experienced with earlier formulations of amphotericin B [89]. Another nanomedicine loaded with ciprofloxacin binds endotoxins using nanostructures to neutralize bacterial LPS [89]. Early animal studies have demonstrated decreased cytokine levels in those administered this therapy. In a murine study, lipid nanomaterials delivering mRNA encoding antimicrobial proteins augmented macrophages’ ability to remove MDR bacteria [110]. This promising technology is expected to yield further advances in sepsis therapeutics.

### 4.2. Host-Directed Therapies

#### 4.2.1. Angiotensin 2

Angiotensin 2, approved by the FDA in 2018, is the newest available vasopressor for the treatment of vasodilatory shock. Angiotensin 2, a naturally occurring hormone in the body, is the end product of the renin–angiotensin–aldosterone system. It causes smooth muscle contraction and releases ADH. In the setting of high dose vasopressors, exogenously administered synthetic angiotensin 2 significantly improved mean arterial pressure (MAP), decreased background vasopressor dose, and lowered sequential organ failure assessment (SOFA) scores in patients with refractory septic shock [111]. In those with an absolute or functional deficiency of angiotensin-converting enzyme, manifested by a ratio of angiotensin 1/angiotensin 2 of ≥ 1.63 or elevated renin levels, angiotensin 2 supplementation statistically significantly improved survival [112,113]. This vasopressor shows promising results in conditions, such as ARDS, influenza, pneumonia, cirrhosis, acute kidney injury requiring renal replacement therapy, respiratory failure requiring veno-venous extracorporeal membrane oxygenation, post-cardiopulmonary bypass, cardiac arrest, and Covid-19-induced shock [114,115,116,117,118,119,120].

#### 4.2.2. Selepressin

Selepressin, a vasopressin analog highly selective for the V1a receptor, targets both V1a and V2 receptors. In experimental studies, selepressin decreases microvascular leakage and increases mean arterial pressure (MAP) at lower doses than vasopressin [121]. A promising phase 2a RCT demonstrated that its use was associated with reduced doses of norepinephrine, less fluid administration, and shorter duration of mechanical ventilation [122]. A subsequent phase 2b/3 RCT confirmed a higher MAP, lower norepinephrine requirement, and lower net fluid balance in the selepressin group, but it did not confirm a difference in ventilator-free days or mortality [90]. Due to its lack of catecholaminergic stimulation, selepressin may be especially helpful in patients with concomitant tachydysrhythmias [85]. This agent is not approved by the US FDA currently.

#### 4.2.3. Mesenchymal Stem Cells

Mesenchymal stem cells (MSCs) can transform to replace damaged or destroyed cells [123]. These pluripotent mesenchymal stem cells have been shown to reduce injury and mortality in animal models of sepsis by restoring endothelial barrier function and enhancing tissue repair [123]. Two ongoing phase 2 trials are evaluating the effect of MSCs on organ failure in patients with septic shock and severe community acquired pneumonia [85,124]. Despite early promising data, concerns exist about the potential for MSCs to become oncogenic, as these cell types have been identified in tumors such as gastric adenocarcinomas, lipomas, and osteosarcomas [125].

#### 4.2.4. Extracellular Vesicles

Extracellular vesicles (EVs) are a group of membrane-enclosed particles released from cells involved in intercellular communication [124]. These vesicles transport RNA and proteins that modulate the immune response of lymphocytes [123]. With similar properties to MSCs, EVs appear to exhibit superior safety [124]. RNA and proteins delivered by EVs play a prominent role in angiogenesis, apoptosis, and immune response while protecting against sepsis-induced organ dysfunction [123]. Murine models highlight its ability to attenuate bacterial pneumonia by enhancing macrophage phagocytosis [125] and improve renal recovery for sepsis-induced acute kidney injury [126]. These findings have translated successfully to patients with chronic kidney disease by improving glomerular filtration rate [127]. Challenges remain in isolating MSC–EVs, and no current standardized protocol for creating or naming EVs exists [124].

#### 4.2.5. Toll-Like Receptor Ligand Binders

Immune cells express toll-like receptors that release cytokines such as tumor necrosis factor-α and IL-6 to induce a strong innate immune response [100]. Binding TLRs could attenuate the immune response in those with sepsis; however, some experts have expressed concern that interfering with this pathway could cause autoimmune, cardiovascular, neurological, and oncogenic disorders [91]. While TLR agonists have demonstrated therapeutic promise in cancer immunotherapy [85], their antagonists have been used to effectively treat polymicrobial sepsis in mice [128]. Though the TLR4 antagonist Eritoran did not show a mortality benefit in patients with severe sepsis, its indirect mechanism of endotoxin inhibition may play a more targeted role in those with elevated endotoxin levels [129]. A direct antagonist anti-TLR4 monoclonal antibody has been developed to treat rheumatoid arthritis and may be a worthwhile target for further investigation in those with sepsis [91].

#### 4.2.6. Interleukin Agonists and Antagonists

Immune modulation using both interleukin agonists and antagonists have been studied in the treatment of sepsis [121].

Although interleukins contribute to host defense against infections [130], exaggerated synthesis of IL-6 can cause an acute severe systemic inflammatory response known as cytokine storm or cytokine-release syndrome [130]. Prior to Covid-19, IL-6 receptor (IL-6R)-targeted agents (e.g., tocilizumab, sarilumab) were used mainly to treat various autoimmune disorders such as rheumatoid arthritis [131]. While recent randomized control trial of tocilizumab have reported favorable responses in patients with Covid-19 pneumonia [132], the data are inconsistent [133], and their therapeutic role for Covid-19 remains unclear. In addition, more than 20 clinical studies are currently registered on ClinicalTrials.gov (accessed on 12 January 2020) that aim to evaluate the efficacy of monoclonal antibodies against IL-6 (e.g., siltuximab, sirukumab, olokizumab, clazakizumab) in Covid-19-induced sepsis and septic shock [93]. Debate surrounds the possible increased infectious risks associated with therapies like IL-6- or IL-6R-targeted agents [134,135]; however, findings from these ongoing studies will expand the understanding of potential clinical applications of IL-6 pathway inhibition to non-Covid-19 sepsis as well.

Interleukin-7 (IL-7) is a cytokine that reverses sepsis-induced lymphopenia, prevents apoptosis, and induces T cell proliferation [136,137,138]. It has been shown to prevent death in animal models with abdominal infection due to *Pseudomonas aeruginosa* [139]. In a phase 2 trial, administration of IL-7 improved absolute lymphocyte count without worsening inflammation in those with septic shock [138].

While administration a recombinant IL-1 receptor antagonist did not improve survival in sepsis, Anakinra, a recombinant IL-1R antagonist which blocks IL-1 release, reduces mortality in septic patients with markedly elevated IL-1RA levels [140,141]. The inability to rapidly assess IL-1 levels limits its broad application to patients with sepsis; however, ongoing phase 2 trials expected to result in the coming year should further elucidate valuable information to be gained on the matter [85].

#### 4.2.7. Cyclic GMP-AMP synthase-stimulator of interferon genes (cGas-STING)

When triggered by PAMP recognition, the cGas-STING pathway in the innate immune system potently activates the inflammatory response [92]. Ceritinib, an FDA-approved drug for use in non-small cell lung cancer, targets this pathway. Despite no current human trials in septic patients, targeting this pathway in murine models of sepsis induced with cecal ligation and puncture demonstrated a survival benefit [142].

#### 4.2.8. Adrenomedullin

Sepsis causes endothelial dysfunction that results in vascular leak, thrombosis, and organ dysfunction [85]. Adrenomedullin counteracts this by stimulating ADM receptors which then maintain the endothelial barrier and decrease inflammation [100,143]. Adrecizumab is a monoclonal antibody that targets anti-ADM antibodies and prolongs the half-life of ADM [97]. Because preclinical trials have shown promise in limiting endothelial damage and the therapy appears safe in phase 1 trials [144], a phase 2 RCT is currently underway [144]. As ADM also causes systemic vasodilation, its modulation in the setting of septic shock may limit its application [100].

#### 4.2.9. Eculizumab

Sepsis-induced complement activation contributes to tissue damage and organ dysfunction [100]. Therapies targeting complement and its uncontrolled activation during sepsis have been successfully studied in baboons and resulted in improvements in defects of coagulation and multisystem organ function [145]. Eculuzimab, an FDA-approved monoclonal Ab targeting C5a for atypical hemolytic uremic syndrome, is currently being studied in phase 2 trials for patients with Covid-19-induced sepsis [146].

#### 4.2.10. Interferon Gamma

Interferon gamma (IFN-ɣ) increases tumor necrosis factor production in patients with sepsis [147]. Though FDA-approved to treat certain malignancies and chronic granulomatous disease by promoting proinflammatory cytokine release, case series documenting IFN-ɣ infusions in patients with fungal sepsis demonstrated improvement in laboratory data and safety [148,149]. Despite preclinical data suggesting a benefit in sepsis, no published RCTs have evaluated IFN-ɣ for this purpose [149].

#### 4.2.11. Triggering Receptor Expressed on Myeloid Cells-1 and Nangibotide

TREM-1 is a receptor expressed on monocytes and neutrophils that, when activated, triggers systemic inflammation [85]. Nangibotide is a TREM-1 antagonist that inhibits the overactive inflammation that can accompany infection [150,151]. In phase 1 trials, nangibotide was found to be safe, though demonstrated a very short half-life of ~3 min [151]. In a follow up phase 2 trial in patients with septic shock, the nangibotide group had decreased SOFA scores with this trend being even more pronounced in those with elevated soluble TREM-1 levels [94].

#### 4.2.12. Immune Checkpoint Modulators

T cells are inhibited when programmed cell death ligands 1 (PD-L1) and 2 (PD-L2) bind the programmed cell death-1 (PD-1) receptor expressed on their surface [85]. Patients with sepsis have increased PD-1 and PD-L1 expression, which correlates with increased mortality [152]. Nivolumab, a monoclonal antibody directed against PD-1 which prevents binding of PD-1/PD-L1, enhances IFN-ɣ production inducing an immune system response to infection [153]. Inhibiting PD-1, PD-L1, and cytotoxic T-lymphocyte antigen-4, a stimulatory molecule that is upregulated and suppresses T cell function in sepsis, improves survival in murine models of fungal sepsis [154]. In phase 1 and 2 trials, nivolumab was shown to be safe and improved absolute lymphocyte count (ALC) in patients with vasopressor-dependent sepsis and low ALC [155].

#### 4.2.13. Granulocyte-Macrophage Colony-Stimulating Factor

Granulocyte-macrophage colony-stimulating factor (GM-CSF) stimulates the production of neutrophils and macrophages in the bone marrow. Administration of GM-CSF promotes immune reconstitution to fight infection and reduce the time to infection resolution [156]. GM-CSF is predominantly used in patients at increased risk of infection due to chemotherapy-induced neutropenia. In an RCT of patients with sepsis-associated immune suppression, GM-CSF administration significantly reduced the length of mechanical ventilation [157].

## 5. Discussion/Conclusions

Many novel and promising diagnostics and therapeutics are being developed to aid in the management of septic patients. Improved diagnostics will permit earlier diagnosis of sepsis, helping to characterize both specific infecting organisms and pathways that become dysfunctional during sepsis. Despite substantial progress in our understanding of the pathophysiology of sepsis, no single sepsis biomarker has yet to address all diagnostic needs. Although combining multiple biomarkers with clinical scoring systems may outperform any single tool, these practices must also be validated in clinical practice. The wide adoption of diagnostic sepsis biomarkers has been hampered by both the lack of a gold standard for diagnosing sepsis and inherent study limitations (e.g., size, design, clinical applicability). Further investigations of appropriate specimens, testing assays, and cutoff levels for specific biomarkers are needed prior to large-scale integration into clinical decision making.

While further evidence is needed prior to widespread adoption of novel therapeutics, their addition to current management has the potential to revolutionize sepsis care. Rapidly targeting antimicrobial therapy to particular pathogens, removing the inciting infectious stimulus, and personalizing therapies according to each individual’s inflammatory response and sepsis phenotype should herald a new era of improved clinical outcomes in patients with sepsis.

## Figures and Tables

**Figure 1 biomedicines-09-00311-f001:**
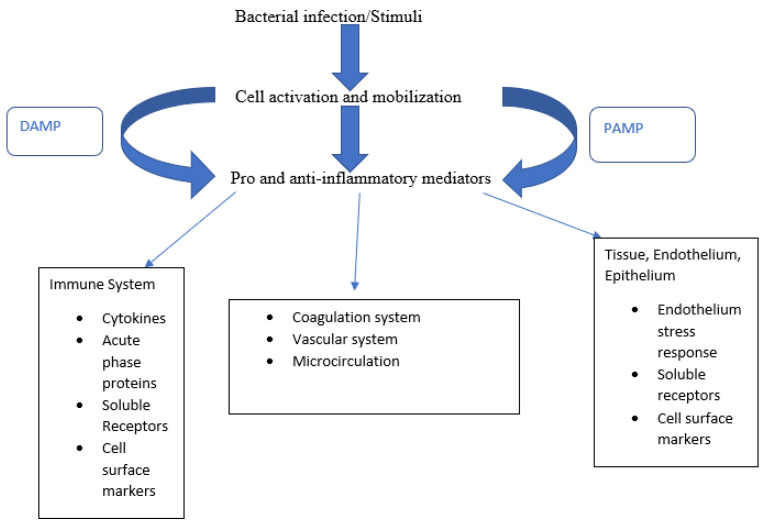
Biomarkers sorted according to their pathophysiological role. Bacterial stimuli cause cell activation and, along with PAMPs and DAMPs, release pro-inflammatory mediators triggering a broad host response.

**Table 1 biomedicines-09-00311-t001:** Summary of biomarkers for novel therapeutics for sepsis.

Summary of Biomarkers	
1. Innate response biomarkers	a. Pathogen-associated molecular patterns (PAMPs)
	b. Damage-associated molecular patterns (DAMPs)
	c. Calprotectin
2. Cytokine/Chemokine biomarkers	a. Interleukin 6 (IL-6)
	b. Monocyte Chemoattractant Protein 1 (MCP1)
	c. Pentraxin (PTX) 3
	d. sTNFR1
3. Receptor Biomarkers	a. Presepsin
	b. CD64
	c. Soluble triggering receptors expressed on myeloid cells (sTREM-1)
	d. TLR-4
	e. PD1
4. Microcirculation related biomarkers	a. Angiopoietin-1 (Ang-1) and Angiopoietin-2 (Ang-2)
	b. Adrenomedullin (ADM) and Pro-Adrenomedullin (ProADM)
5. Biomarkers of Organ Dysfunction	a. Micro-RNA (miRNA)
	b. Long Non-Coding RNAs (LncRNAs)
	c. Matrix Metalloproteinases (MMPs)

**Table 2 biomedicines-09-00311-t002:** Summary of benefits, concerns, and current phase of clinical trials for novel therapeutics for sepsis.

Therapy	Benefit	Concern	Phase of Clinical Trial
PAMP Removal	Improved hemodynamics; improved mortality in murine model	Differing mechanisms/targets of removal between devices. No studies assessing effect on mortality to date	Emergency Food and Drug Administration (FDA)-approval for Covid-19, ongoing multicenter clinical trials [83]
Bacteriophages	Can neutralize multidrug-resistant (MDR) bacteria	No randomized controlled data assessing efficacy	Case reports in humans [84]
Intravenous immunoglobulin (IVIG)	Useful in certain inflammatory conditions	No defined benefit in sepsis patients	FDA-approved for immunodeficiencies and inflammatory conditions
Targeted Monoclonal Antibodies	Avoids antibiotics resistance	Each drug only effective against targeted organism	Phase 3 trials underway [85]
Liposomes	Can bind bacterial toxin to minimize damage	Limited use in bacteria that secrete endotoxin	Phase 1 trials completed [86]
Alkaline Phosphatase	Mortality reduction in septic shock with acute kidney injury	Benefit found in only those with acute kidney injury	Phase 2 trials [87]
Antimicrobial Peptides	Synergism with antimicrobials	Cytotoxicity towards host cells	Phase 3 trials [88]
Nanoparticles	Increase potency and minimize side effects of antimicrobials	High development costs	Liposomal amphotericin B FDA-approved [89]
Angiotensin II	Catecholamine-sparing effect; improved mortality in certain patient populations	Limited prospective experience outside of phase III trials	FDA-approved for use in septic shock
Selepressin	Catecholamine-sparing effect with lower net fluid balance	No change in ventilator/vasopressor-free days	Phase 3 trial completed [90]
Mesenchymal Stem Cells	Decreased cell injury in murine sepsis models	Concern for oncogenicity	Phase 2 trials [85]
Extracellular Vesicles	Shown to improve renal recovery in murine models of sepsis	No standard nomenclature/isolation techniques	Phase 2 trials [91]
TLR4 Ligand Binders	Positive results in murine models of sepsis	Potentially oncogenic	FDA-approved only in the setting of cancer therapy
Interleukin agonists/antagonists	IL-7 agonist: prevents lymphopenia in septic shock; Anakinra: improved mortality in those with elevated IL-1RA levels; IL-6R and IL-6 antagonist: attenuates cytokine storm	IL-7 agonist: No mortality benefit in current trials; Anakinra: No data for routine use in sepsis IL-6R and IL-6 antagonist: mixed data, no data for non-covid sepsis	Phase 2 trials [92]; Anakinra FDA-approved for rheumatoid arthritis IL-6R and IL-6 antagonist: phase 2 and phase 3 trials [93]; FDA-approved for rheumatoid arthritis, EUA for Covid-19
cGAS-STING (cyclic GMP-AMP synthase-stimulator of interferon genes)	Murine models of sepsis demonstrated survival benefit	No in human data to suggest benefit in sepsis	FDA-approved for non-small lung cancer
Adrenomedullin	Potential to decrease capillary permeability in sepsis	Concern with potential of hypotension	Phase 2 trials [85]
Eculizumab	Improved multiorgan dysfunction in Baboon models of sepsis	May lead to immunosuppression	FDA-approved for use in atypical hemolytic uremic syndrome
Interferon Gamma	Case series demonstrating improved cytokine profile	No RCT studying IFN-ɣ in sepsis	FDA-approved for chronic granulomatous disease and certain malignancies
Soluble TREM-1 and Nangibotide	Improved SOFA scores, especially in those with elevated sTREM-1 levels	Short half-life requires infusion	Phase 2 trials [94]
Immune Checkpoint Modulators	Improved absolute lymphocyte count (ALC) in those with low ALC and septic shock	Patient relevant clinical outcomes unknown	Phase 2 trials [92]
Granulocyte-Macrophage Colony-Stimulating Factor (GM-CSF)	Reduced length of mechanical ventilation for sepsis-induced immunosuppression	No clear mortality benefit in sepsis	FDA-approved for chemotherapy-induced neutropenia

## Data Availability

No new data were created or analyzed in this study. Data sharing is not applicable to this article.

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
