# Peer review of "Novel Diagnostics and Therapeutics in Sepsis"

_biomedicines, 2021, doi:10.3390/biomedicines9030311_

Round 1

Reviewer 1 Report

The article by Leong et al. is an important review article that people in the field will find useful! The categorizing of biomarkers with regard to different sepsis-related pathways is interesting! The categories are well chosen. I have a few minor criticisms as listed below!

line 24: If I am not mistaken Paoli et al are referring to health care costs in 2013? Please check the reference!

line 70, 71: the use  of MALDI-TOF is mentioned, and cited with references 9/10. Ref 10 talks about MALDI-TOF, referencing 9, while ref. 9 does not mention MALDI-TOF. It mentions mass spectrometry in general in the abstract, while describing Raman spectroscopy in the article. The origin of 4 hours in ref 10 is also unclear, since the Raman spectroscopy was apparently done on blood cultures!

line 90: should it read reduced mortality?

line 112: I find the reference to immobilized Fc-MBL in the context confusing, since this is not a natural way of pathogen recognition! It should probably be introduced in line 115 with the blood assay.

line 148: It was my understanding that IL-6 is mainly elevated in bacterial infections, and COVID-19 is somewhat of a unique example there? Thus I don’t understand why IL-6 levels in viral infections are less inconclusive than in bacterial infections? (e.g. doi:10.1371/journal.ppat.1005973)

line 226: the technique is interesting, but mainly seems based on a microfluidic chip, rather than on a smartphone. The latter merely seems to act as a camera that can be plugged into a specifically made microscope!

line 246-257: I do not see the relevance of TLR-4 (and other TLRs) as a biomarker, and I do not see that this is established/discussed here either. Of course the functional TLR4 relevance for LPS detection is indisputable!

line 306/307 is not a sentence

line 498/499: for the effective treatment of polymicrobial sepsis in mice a review article on computational methods is cited

Reviewer 2 Report

The authors provide an extensive summary of biomarker and treatment options in sepsis. They have included the vast majority of options, especially treatments with potential interest in sepsis. Few things beyond the already listed therapies could be added in the manuscript. I would just like to do a few comments that to make it easier for potential readers.

Major comments:

  • The vast majority of biomarkers also have their counterpart as a therapeutic option. Defining and describing them the first time in the diagnostic part is enough. It should not be done again a few pages later in the therapy section. That makes the review repetitive and excessively long. E.g. lines 393-398 for defining PAMPs which have already been introduced in the biomarker section.
  • The therapeutics section should be divided into two subsections: 1. Pathogen-directed therapies. 2. Host-directed therapies. I think this could make it easier to read. The current order is a bit hodgepodge and chaotic.
  • The treatment options "4.1.1. Pathogen-associated molecular pattern removal devices", "4.2.5. Toll-like receptor ligand binders" and "4.2.6. Interleukin agonists and antagonists" have failed clinical trials in the past including AntiEndotoxin (Angus DC, Birmingham MC, Balk RA, et al. E5 murine monoclonal antiendotoxin antibody in gram-negative sepsis: a randomized controlled trial. E5 Study Investigators. JAMA 2000;283(13):1723–30. ), AntiTLR4 (Opal SM, Laterre P-F, Francois B, et al. Effect of eritoran, an antagonist of MD2-TLR4, on mortality in patients with severe sepsis: the ACCESS randomized trial. JAMA 2013;309(11):1154–62. ), AntiTNF (Fisher CJ, Agosti JM, Opal SM, et al. Treatment of septic shock with the tumor necrosis factor receptor:Fc fusion protein. The Soluble TNF Receptor Sepsis Study Group. N Engl J Med 1996;334(26):1697–702. ) and AntiIL-1R (Fisher CJ, Dhainaut JF, Opal SM, et al. Recombinant human interleukin 1 receptor antagonist in the treatment of patients with sepsis syndrome. Results from a randomized, double-blind, placebo-controlled trial. Phase III rhIL-1ra Sepsis Syndrome Study Group. JAMA 1994;271(23):1836–43.). The authors should explain what has changed since then and why now they should have a new chance. One possible reason could be the better classification of patients nowadays and the possibility of improved treatment personalization. In the same line of the example they cite from Anakinra (it shows efficacy only in those with elevated IL-1RA). This point must be discussed.
  • PD-L1 and Immune-checkpoints. In point 3.3.6 the ligand, PD-L1, , must be also added as severity marker (Avendaño-Ortiz J, Maroun-Eid C, Martín-Quirós A, et al. Front Immunol 2018;9:2008) and (Wilson JK, Zhao Y, Singer M, Spencer J, Shankar-Hari M. Crit Care 2018;22(1):95.). In therapeutics I recommend change "4.2.12 Programmed Cell Death Protein 1 (PD-1) and Programmed Cell Death 1 Ligand 1 Antibodies" to "Immune checkpoints" in general as other such as CTLA-4 have been also postulated (Chang KC, Burnham C-A, Compton SM, et al. Crit Care 2013;17(3):R85.) and (Wykes MN, Lewin SR. Nat Rev Immunol 2018;18(2):91–104).

Minor points

  • Line 68. "Several molecular approaches have been developed in order to improve ...."
  • Figure 1 may be improved. Curved arrows point to nowhere. The tabs in the text are messy and aesthetic.
  • Lines 72-77. A 5-line sentence with just a comma is too much. Please consider fragment it.
  • Confidence Interval acronym appears in line 127 for first time. However, its acronym is defined in line 165 and another time in 196. Please revise the acronyms.
